# Reinforcement Learning to Send Reminders at Right Moments in Smartphone Exercise Application: A Feasibility Study

**DOI:** 10.3390/ijerph18116059

**Published:** 2021-06-04

**Authors:** Shihan Wang, Karlijn Sporrel, Herke van Hoof, Monique Simons, Rémi D. D. de Boer, Dick Ettema, Nicky Nibbeling, Marije Deutekom, Ben Kröse

**Affiliations:** 1Informatics Institute, University of Amsterdam, 1090 GH Amsterdam, The Netherlands; h.c.vanhoof@uva.nl (H.v.H.); b.j.a.krose@hva.nl (B.K.); 2Department of Information and Computing Sciences, Utrecht University, 3584 CC Utrecht, The Netherlands; 3Department of Human Geography and Spatial Planning, Utrecht University, 3584 CS Utrecht, The Netherlands; k.sporrel@uu.nl (K.S.); d.f.ettema@uu.nl (D.E.); 4Consumption & Healthy Lifestyles Group, Wageningen University & Research, 6700 HB Wageningen, The Netherlands; monique.simons@wur.nl; 5Digital Life Centre, Amsterdam University of Applied Science, 1091 GC Amsterdam, The Netherlands; r.d.d.de.boer@hva.nl; 6Centre of Expertise Urban Vitality, Amsterdam University of Applied Science, 1097 DZ Amsterdam, The Netherlands; n.nibbeling@hva.nl; 7Faculty of Health, Sports and Welfare, Inholland University of Applied Sciences, 2015 CE Haarlem, The Netherlands; marije.deutekom@inholland.nl

**Keywords:** reinforcement learning, just-in-time adaptive intervention, reminder, physical activity, mobile application

## Abstract

Just-in-time adaptive intervention (JITAI) has gained attention recently and previous studies have indicated that it is an effective strategy in the field of mobile healthcare intervention. Identifying the right moment for the intervention is a crucial component. In this paper the reinforcement learning (RL) technique has been used in a smartphone exercise application to promote physical activity. This RL model determines the ‘right’ time to deliver a restricted number of notifications adaptively, with respect to users’ temporary context information (i.e., time and calendar). A four-week trial study was conducted to examine the feasibility of our model with real target users. JITAI reminders were sent by the RL model in the fourth week of the intervention, while the participants could only access the app’s other functionalities during the first 3 weeks. Eleven target users registered for this study, and the data from 7 participants using the application for 4 weeks and receiving the intervening reminders were analyzed. Not only were the reaction behaviors of users after receiving the reminders analyzed from the application data, but the user experience with the reminders was also explored in a questionnaire and exit interviews. The results show that 83.3% reminders sent at adaptive moments were able to elicit user reaction within 50 min, and 66.7% of physical activities in the intervention week were performed within 5 h of the delivery of a reminder. Our findings indicated the usability of the RL model, while the timing of the moments to deliver reminders can be further improved based on lessons learned.

## 1. Introduction

Inactive lifestyle and a lack of physical activity can lead to serious physical and mental health issues [1]. People are therefore advised to engage in regular physical activity (i.e., at least 150 min of moderate-intensity activity every week for adults [2]). However, many individuals struggle to maintain a healthy activity level. Consequently, the promotion of healthy physical activity behavior is still an open challenge [3]. Interventions using mobile exercise applications are considered promising for supporting physical activity behaviors, as mobile devices are well integrated into people’s daily lives [4] and can continuously deliver interventions to users [5].

Recently, just-in-time adaptive intervention (JITAI) has gained attention and been considered to be an effective strategy in the field of mobile healthcare intervention [6]. JITAI is an intervention design aiming to provide the right type and amount of support, at the right time, by adapting to an individual’s changing internal and external state [7]. In the JITAI framework, identifying the right moment for intervention is a crucial component. As argued by Fogg, in daily life, the personal state that the user is involved in when an intervention is delivered can play an important role in determining its effectiveness [8]. For instance, a reminder about physical activity sent during a meeting will likely be ineffective, no matter how persuasive its content might be.

JITAIs in mobile health applications are mainly used to prevent certain health threats, including addictive behaviors such as overeating [9,10], smoking [11] and prolonged sedentary behaviors [12]. In other words, that research mainly concentrates on finding ‘vulnerable states’ in which a user is urged not to perform a certain behavior. A few recent systems have started to research the right moments of ‘opportunity’ to promote a certain behavior, although they are still at the early stage. Ding et al. [13] described the potential of such a system and demonstrated how sending reminders at the right moments could improve the persuasiveness of mobile interventions. In their study, researchers set predetermined opportunistic moments in which the user is invited to walk (e.g., when the individual is overusing a smart phone or when he or she is sedentary). Expanding on this previous research, we proposed the incorporation of a data-driven ‘moment of opportunity’ in a smartphone application to promote user physical activity.

However, identifying the ‘opportunity’ moments from data to deliver interventions in mobile applications is demanding, as the right moments for an individual can change according to personal momentary needs. One possible solution is to use the reinforcement learning (RL) technique, as it can continuously adapt the intervention strategy based on both the momentary state and the feedback of users [14]. RL has been used in mobile applications to support a certain physical activity. Most researchers have continuously adapted the *content or type* of interventions by tracking the momentary behaviors of people. For instance, Rabbi et al. [5] recommended different types of physical activity adapting to users’ changing needs using a bandit algorithm. Yom-Tov et al. [15] personalized the contents of messages for patients with diabetes to encourage their physical activity. Zhou et al. [16] delivered interventions to users suggesting adaptive and personalized daily step goals. In an intervention focused on weight loss, Forman et al. [17] optimized the combination of text messages and human coaching intervention based on a cumulative reward score. Conversely, several researchers have concentrated on using RL to adapt to the *moments* of intervention delivery with respect to users’ on-going status and behavior. In one recent work, Liao et al. [18] developed the HeartSteps application to actively decide whether to provide interventions at five fixed moments per day. However, too many interactions over a short period (e.g., 5 times per day) could add burden for the user and adversely impact engagement [19]. Driven by this practical issue, we allow a limited number of reminders within each week in our smartphone application, and use RL to identify the ‘right’ moments for sending a restricted number of reminders. As far as we know, this has never been studied in mobile health intervention settings.

In this paper, we concentrate on motivating users to perform a physical activity by delivering reminders at personalized moments. The moments when reminders are sent depend on the personal momentary context of users (i.e., time and calendar). To examine the usability of such an RL-based smartphone exercise application, a feasibility study with real target users was conducted and presented. We aim to explore both the user behavior (i.e., interactions with the reminders) and the user experience after receiving reminders delivered at the personalized moments. Thus, the following three hypotheses were addressed and examined:Reminders sent at personalized moments can trigger active reactions within a short period of time.Reminders sent at personalized moments can trigger users to start a physical activity.Reminders sent at personalized moments are perceived as useful by users, based on user feedback from questionnaires and interviews.

## 2. Materials and Methods

To examine the proposed hypotheses, we first designed and integrated an RL model in a smartphone exercise application to deliver reminders based on users’ momentary temporal context. Then, we conducted a feasibility study, in which 11 target users participated. At the end, we collected and analyzed data from 7 users who used the smartphone exercise application for 4 weeks and received reminders sent at moments determined by the RL model. The 7 users were also asked to fill out a questionnaire and have an interview to share their user experience.

### 2.1. Materials

#### 2.1.1. A Smartphone Exercise Application Based on Reinforcement Learning

In our study, a smartphone exercise application (the PAUL app) was developed to send reminders adaptively at moments suggested by a reinforcement learning (RL) model. The development of the PAUL app was conducted based on both theoretical foundations of behavior change [20] and a focus-group study with our target users [21]. The design process and practical implementation of this smartphone exercise application can be found in [22]. In this subsection, we mainly describe the integration of a reinforcement learning model in the PAUL app.

In Figure 1, we present an overview of how the PAUL app iteratively interacts with the user to decide the right moments to deliver reminders, where the RL model is marked in green. In this interaction procedure, the RL model is a decision-maker in the smartphone exercise app. On the one hand, the RL-based decision-maker determines the momentary status of a user based on personal context information collected by the app at each time step, then estimates whether this is the right moment to send a reminder to this user. Furthermore, the RL-based decision-maker records the physical activity behavior of the users after receiving the reminders. In this manner, the RL model learns the preference of this user from his or her historical data, and keeps the state-of-the-art knowledge to make the next decision.

We decided that the PAUL app should decide on whether to send a reminder or not at every hour from 8:00 to 20:00 in each day. Additionally, since too-frequent interactions with the user are not desirable in our practical task, we set the maximum number of reminders sent. In general, the PAUL app sends a maximum of 14 reminders each week. The content of the messages were drafted based on previous literature [23] and evaluated in a pilot test. Based on the results of questionnaires, a message library of 141 messages was built. The messages are positively framed, focus on effectiveness and immediate outcomes, and are tailored to the activity type (i.e., running, walking or both).

#### 2.1.2. The Reinforcement Learning Model

We modeled our practical task (i.e., how to learn the optimal strategy for delivering reminders with respect to the momentary context of this user) as a reinforcement learning problem. For each user, we formalized the problem as a Markov decision process (MDP). The MDP framework is a mathematical abstraction of sequential decision-making problems [14]. Figure 2 depicts the interaction between an agent and an environment in our MDP. Here, the agent represents a mobile health system that learns the optimal strategy to interact with a target user, which is our environment. Our agent and environment interact in a sequence of discrete and finite time-steps {1,2,…,t}, which can be naturally broken into episodes. At each time step, the agent observes the contextual representation of the environment, and on that basis selects an action. Afterwards, the environment passes a numerical reward (inferring its feedback on the given action) back to the agent, and updates itself in a new state. Based on this trial-and-error mechanism, the agent adapts its policy for maximizing an expected long-term reward. We follow the standard definition in [14] and define the key elements of our MDP as follows:*S*: A set of observed *states*, each state $s_{t}\in S$ is a vector of contextual variables of the environment (i.e., user), including ‘the number of reminders left in this week’, ‘timestep from the last run’, ‘timestep from the last reminder’ and ‘hour of the day’, ‘weekday’, ‘calendar availability’.*A*: A set of *actions* taken by the agent (i.e., app), two actions, ‘sending reminder’ and ‘not sending reminder’, are denoted as $a_{t}=1$ and $a_{t}=0$ respectively.*T*: A probabilistic transition function, defining the probability of moving from $s_{t-1}$ to $s_{t}$ given a specific action $a_{t-1}$.*R*: A *reward* function, representing the probability that a reward $r_{t}$ was received by the agent. If the target user performs a physical activity at time *t*, the reward $r_{t}=1$; if not, $r_{t}=0$.

In our case, we aim to optimize the frequency of physical activities in a week with the PAUL app. Following the setup of sending reminders in the PAUL app, the RL model decides on which action (sending reminder or not) to take at every hour from 8:00 to 20:00 in each day. In total, there are 84 decision time-steps in each week. If a physical activity is performed by the user before the next decision time step (within one hour), we gave a reward of 1.0 to our app (otherwise zero reward is assigned). Since we allow a constrained number of reminders, our optimization goal becomes how to wisely deliver the maximum number of reminders at the ‘right moments’ in an episode to maximize the user’s physical activity frequency in that period.

#### 2.1.3. The Pre-Learned Reinforcement Learning

In this section, we explain one important property of our RL model. A problem that many RL-based mobile intervention systems suffer from is the cold-start problem [24]. As very few (or even no) experiences with the user are available at the beginning, RL models often require the application to interact many times with the users prior to performing well (learning how to deliver the reminders at the right moments in our case). To remedy this problem, we pre-learned a generalized initial delivery strategy in our RL model from the large-scale empirical running data. An overall framework of our RL smartphone exercise application is shown in Figure 3.

Following this procedure, we first clustered users from a large-scale historical running data and identified the target users [25]. The used running data contains around 440,000 runs over 4 years performed by more than 10,000 users. In the data analysis, we characterized the target users as people who struggled with maintaining running behavior (it is consistent with our prioritized participants in the feasibility study). Moreover, as each data point includes the timestamp and weather information at the start moment for every run, we also captured when and under which weather conditions our target users prefer to start runs. Based on those preferences, we then developed an RL algorithm to learn an initial strategy using this empirical data in simulations [26]. In this manner, we aim to obtain a generalized strategy that delivers a limited number of reminders to our target users at reasonable moments (better than a random initialized one in code-start RL models).

### 2.2. Methods

#### 2.2.1. Study Design

The feasibility study consisted of three steps, namely enrollment, intervention and feedback. First, in the enrollment step, we recruited the participants, visited all eligible participants and asked them to fill out a questionnaire to assess their socio-cognitive characteristics. Second, our participants used the PAUL smartphone application in their own smartphone from 18 November to 15 December 2019. They were asked to turn on the reminder function of their smartphones (i.e., to receive pop-up notifications) and to give access to their digital calendar (i.e., to provide calendar availability data, but they did not have to). We separated this 4-week study period into 2 phases: baseline (3 weeks) and intervention (1 week). In the first 3 weeks, participants had access to the PAUL app, but no JITAI reminders were sent by the app. These initial three weeks without the JITAI reminders allowed us to ensure the usability of basic functions in the app and to collect the baseline behaviors of participants. Then, in the final intervention week, the PAUL app delivered a maximum of 14 reminders to each participant, asking them to perform a physical activity (either a walk or a run). The moments of delivery were decided by the RL model based on the user’s personal momentary information (i.e., time preference and calendar availability). During the whole period, all participants were encouraged (but not obliged) to use the PAUL app as much as possible, and several data on participants’ behaviors (e.g., interactions with the app) were monitored and collected automatically by the PAUL app. Third, at the end of the study, all participants answered a questionnaire on the usability of the reminder function in the PAUL app. In this feedback step, we also conducted a one-to-one interview (20–40 min) with each participant, to understand their experience and feelings on using the PAUL app in the manner we designed.

#### 2.2.2. Participant Recruitment

After being granted approval from the local ethics review board (No. ERB Review Geo S-19253), we started our study by recruiting participants. To maintain a healthy lifestyle, people are advised to engage in a sufficient amount of physical activity on a regular basis. Therefore, our prioritized participants are people who are struggling with maintaining a healthy activity level and would like to participate in more physical activities. During the recruitment, we also accepted participants who perform physical activity regularly but still need to move more to reach the suggested amount.

To control the influence of living location on user behavior, we selected three parks in city centers of the Netherlands (i.e., Transwijk at Utrecht, Oosterpark at Amsterdam and Sloterplas at Amsterdam) and only recruited participants who live close (less than 5 km and 20 min bike ride) to either park. We built up a website to introduce the relevant information of our study for interested individuals and to include an online eligibility questionnaire, which is used for assessing their stage of healthy behavior change (as defined by the Transtheoretical Model (TTM) [27]). Our stakeholders in the project also aided with the recruitment by contacting neighborhood organizations (e.g., promotion materials such as flyers were placed in public areas around the parks). We also recruited participants via Facebook advertisements. Based on the participant applications, we enrolled 11 participants who were aged between 18 and 55 years and met the following criteria: not meeting the physical activity guidelines of 150 min per week, with no medical condition preventing them from performing PA (defined by PAR-Q [28]), owning an Android smartphone, not currently participating in another PA or health-related intervention and have proficient knowledge of the Dutch language (as the app is in Dutch). In total, 7 of 11 participants completed the whole study and received reminders sent at personalized moments by the RL model, whose demographic information is presented in Table 1.

#### 2.2.3. Outcome Measurements, Questionnaire and Interview

During the study, our PAUL smartphone app automatically tracked the physical activities performed by the participants and recorded their reactions after receiving reminders. On the one hand, the data collection of physical activity starts when the user clicks the ‘start exercise’ button of the app and then continuously tracks all the data involved in a run in the background, until it is terminated by the user in a comparable manner. For each physical activity, a dataset is collected summarizing total distance and run time, and marking the timestamp and GPS location at the start point of this physical activity. On the other hand, the data collection of users’ active reactions to reminders starts when they give positive feedback. The positive feedback can be given by either tapping on the pop-up notification corresponding to the reminder or opening the application in a short period of time. Otherwise, we consider users ignore the reminders because they were not sent at the right moments. Based on the collected data, we address the following two outcome measurements.

**The number of reminders actively reacted by a user**: a user was considered to ‘actively react to a reminder’ if he or she taped on the pop-up notification after this reminder was delivered.**The number of physical activities triggered by a reminder**: a physical activity was defined as ‘triggered by a reminder’ if the user performed a physical activity after he or she actively reacted to a delivered reminder.

Using those measurements, we aim to quantify user behavior after receiving our designed reminders and examine the first two hypotheses in Section 1.

In addition, to explore the experience of users on receiving the reminders and to answer the third hypothesis, we designed several questions for a questionnaire and an individual interview based on previous studies [13,29,30]. These questions also concentrate on the receptivity of reminders, which is a concept that anticipates a subjective overall reaction of the user to a notification [30]. We particularly measure two underlying factors of receptivity, which are the willingness to be interrupted (known as reachability) [29] and the likelihood of influencing the recipient’s future actions (known as actionability) [30]. All interviews were audio-recorded and transcribed. In Table 2, we present the questions asked in the questionnaire and the interview respectively (the original ones are all in Dutch). Among them, all questionnaire questions are multiple choices with a 4- or 5-point Likert scale (see the scale details in Section 3.2).

#### 2.2.4. Data Analysis

In this study, we mainly used descriptive analysis and analyzed the data collected from three resources (including the PAUL application, a questionnaire, and interviews). We not only statistically described user objective behaviors from the PAUL application data, but also conducted the descriptive analysis to understand user experience from the questionnaire and interview data.

Regarding the PAUL application data, two datasets were collected and analyzed. On the one hand, a log dataset collected the app usage events of every participant and particularly marked the timestamp of those events. Table 3 gives examples of our collected data in the log dataset. Following two defined outcome measurements in Section 2.2.3 (i.e., reminders actively reacted to by a user and physical activity triggered by a reminder), we tracked the objective behaviors of users after receiving reminders from the PAUL application. On the other hand, a session dataset contained detailed information about all physical activity sessions (i.e., user ID, starting timestamp, duration, length, etc.).

During this process, to make sure the data analysis is valid, we confirmed the consistency of collected data. For instance, a walking or running session data entity was included in the analysis only if its starting timestamp was marked as the same in both datasets from the PAUL app. Additionally, to determine the reliability of the user experience data, we conducted a cross-check between answers in the questionnaire and interviews of same participants.

## 3. Results

In this section, we present the results of the usability of our RL-based application, including user reaction times and physical activity behaviors, as well as their perceived feedback.

### 3.1. Results Related to User’s Objective Behaviors

In this section, we present the quantitative results based on the quantified outcome measurements defined in Section 2.2.3, which are mainly related to users’ objective behaviors.

First, we summarized the physical activities performed by our participants, where the records with less than 60 s were excluded (i.e., the duration of run or walk is less than one minute). We realized that 6 out of 20 physical activities performed by all 7 users were in the intervention week (i.e., accounting for 30%), which is higher than the average in all four weeks. Meanwhile, in total, 79 reminders were sent to our participants in the intervention week. On average, each user received 11.3 reminders per week.

Meanwhile, we present the results of two kinds of reactions performed by all users in Figure 4. We observed that participants tend to behave in a diverse manner after receiving the designed reminders. For instance, user No. 3 actively clicked on most of the pop-up reminders sent to him or her, while none of them transferred into an actual physical activity performance. Conversely, although user Nos. 2, 4 and 7 only reacted to a few reminders (by clicking them), they did seem to be triggered by those reminders and performed physical activities afterwards. To demonstrate the detailed behavior of participants, we present the behavior of user No. 2 during the intervention week in Figure 5 (who was the most active participant in our study).

In addition, by summarizing the data of our participants, we noticed several common behaviors. First, we observed that 4 out of 6 physical activities in this week were performed within 5 h of the delivery of a reminder (accounting for about 66.7%). Those 4 physical activities were contributed by 3 different participants (i.e., user Nos. 2, 4 and 7). Interestingly, user No. 7 only ran one time during the whole feasibility study, which was triggered by our reminder. Second, we found that 18 out of 79 reminders received a reaction from our participants by clicking on the pop-up notification (accounting for 22.8%). Importantly, every participant reacted to at least one reminder. The distribution of the interval time between the reaction (timestamp to click a reminder) and delivery of a reminder is calculated and given in Figure 6. From the figure, we can observe that most reminders (83.3%) were actively reacted to by our participants within 50 min.

### 3.2. Results of Questionnaire and Interview

In this section, we present the qualitative results based on a questionnaire and an interview. We separate the detailed results of the questionnaire in Figure 7 (Q1–Q6 related to the timing of the reminders) and Table 4 (Q7–Q9 related to other subjective experience of the reminders), while we jointly discuss the findings from the questionnaire and the interview.

In summary, all participants indicated that it was very important to receive reminders to initiate a running or walking session. Some explained that reminders ‘confront’ them with not doing the exercise and reaching their goal (user No. 5) or that they would otherwise forget to go for a run or walk (user No. 7) or to use the application (user No. 3). One of the participants also claimed that although the reminders of applications sometimes annoyed her, she did think they are useful, because you would not download such an application if you do not need to be pushed:


*“...no signal at all is not a kick in the ass, and apparently you need it, that’s why you need an app, yeah, and you install it”*
(user No. 2)

Furthermore, our participants also highlighted that the right ‘timing’ of reminders is significant to make sure they are effective. If the messages are continuously sent at times that people cannot engage in physical activity, they can end up feeling annoyed, or even feel discouraged and disappointed in themselves for not being able to do more activity, while they do want to do it:


*“And every time you think, “Now I can’t, now I can’t.” [sighs] And that feels like giving up. […] That’s just demotivating for, um, using the app very much. And not necessarily for moving. That you think after a while: “Oh I remove the app, because this is not nice anymore”. Like that. I didn’t have that yet, but it can happen.”*
(user No. 7)

According to Q4 in Figure 7, more than half of the participants (n = 4) noticed most reminders immediately after their delivery (including 2 participants almost always noticed them). This result is consistent with the quantified results of number of reminders actively reacted by users shown in Figure 6, which indicates that our reminders were sent at the right timing of ‘capturing the immediate attention of a user’.

In addition, we found that three participants did not perceive the reminders to be sent at the right times (see Q1 in Figure 7). Two of them also stated that there were too few messages (one participant even claimed that he or she never received any reminders), which might explain their negative feedback about the delivery timing. In other words, there is a possibility that the reminders were not sent at times they needed them (e.g., in the first 3 weeks or as frequent as they wanted), instead of at bad times. We also noticed that our reminders did not always motivate people to perform a physical activity. Only about 57% of participants (n = 4) perceived the reminders can motivate them to engage in physical activity (see Q2 in Figure 7), which is similar to the number of participants who actually performed a physical activity during the intervention week (n = 3) (see Figure 4). Interestingly, some participants discussed the reasons that they thought the reminders were not sent in good times in the interview. One important factor is the weather. Since the Dutch weather tended not to be attractive during our feasibility study (e.g., windy, rainy and short daytime), a few participants claimed that they were not very motivated to perform in such weather conditions.

Most participants did not feel annoyed (n = 4) or interrupted (n = 4) by the reminders (see Q5 and Q6 in Figure 7), while they perceived the reminder function (e.g., only one participant turned off the reminders for a couple of days in Table 4). Thus, although the reminders were not always sent at optimal times, they also did not frustrate people to a degree that would result in stopping using the application or perform physical activity.

During the interviews, different topics arose regarding the frequency of the reminders. In general, it seems the preferred number of reminders is related to the goal of the participants. Some participants want to plan their activities and receive a certain number of reminders based on their own settings. However, other people, who do not like to plan their activities, prefer to have reminders daily, so they can decide at that point of time whether they feel like running or walking. See the following answers from the participants:


*“Well, if I set a goal of three times a week, I want a reminder three times a week. And that I can choose on which days, so I can schedule it in advance.”*
(user No. 5)


*“Um, I think one every day, because otherwise a day will soon go by. Say with one of those apps, I’ll miss it quickly. So if there is such a reminder, I thought “oh yeah, I should do that too”. And if you don’t, after all, chances are I won’t do anything about it that day. And if I got that reminder, I’m more likely to think ’oh yes, I had this as my goal, I have to do this for a while’.”*
(user No. 3)

## 4. Discussion and Conclusions

In this paper, we follow just-in-time adaptive intervention (JITAI) and propose to send reminders at adaptive times based on users’ momentary temporal and calendar information. We first integrated the reinforcement learning approach in a mobile exercise application for determining the right moments to deliver a reminder. Then, we conducted a 4-week feasibility study and evaluated the usability of such a reminder delivery method. This study demonstrates that sending reminders at personalized moments through a RL-based mobile application is feasible. These JITAI reminders were sent at times that the participants appear to be receptive, but not always at times that the participants had the opportunity to exercise.

First, the results shown in Section 3 provided positive evidence for the hypotheses. We found that reminders sent in our personalized moments were able to attract user reactions within a short period of time. For instance, all participants actively reacted to the reminders and over 83.3% reactions were done within 50 min of receiving the reminders. This is an important property to increase the reachability of reminders [31] (introduced in Section 2.2.3), as it can respect the user’s willingness of being interrupted. For instance, if a participant is busy with something, it is hard to interrupt him and let him receive the content of this reminder. Meanwhile, reminders sent in our personalized moments could trigger users to start a physical activity. For instance, an inactive participant only performed one physical activity during the entire feasibility study, which was triggered by our reminder. In the interviews, our participants also highlighted that the right ‘timing’ of reminders are significant to make sure they are effective. Those findings are in line with earlier research that indicated the effectiveness of delivery ‘timing’ in mobile interventions [7,8].

Several interesting scientific contributions have also been made in our study. First, it is among the first studies to examine the relationship between the reminder time, the reaction time and the action time (actual performance time) for physical activity interventions. For instance, according to the questionnaire, most participants would like to have about 1 h between the reminder and the actual exercise. In addition, shown in the behavioral results, the reaction time and action time were about 1 h and 5 h respectively. As the reaction and action behaviors could reflect the reachability and actionability of interventions respectively [29,30], we think those two interval times (action time and reaction time) shall be further explored in determining the effectiveness of mobile health interventions. Second, as far as we know, our RL method incorporated a constraint on interaction frequency (i.e., the maximum number of reminders per week) for the first time in RL-based mobile intervention applications. According to the questionnaires and interviews, only 2 out of 7 users expressed that there were too many reminders. Additionally, most of participants rarely felt annoyed or interrupted by the reminders. As the frequency of interactions could add burden for the user engagement [19], we therefore recommend that future applications also restrict the frequency of reminders when developing RL methods for mobile health interventions. Furthermore, we generated results from both quantified outcome measurements and qualitative studies (i.e., questionnaire and interview). By comparing those results, we noticed an interesting phenomenon. A participant claimed that he or she did not receive any reminder in the interview (in Section 3.2); however, based on our data collection, all participants have clicked on the notification of a reminder at least once (see Figure 4). We think this phenomenon was caused by the deviation between physical behavior and awareness of the participant. As discussed in a recent review, the automatic detection from sensing data could reduce the burden of self-reporting [32]. This finding indicated the importance of combining results from the non-invasive methods with the qualitative methods in feasibility studies.

Besides the interesting findings, there are still several limitations to our research. First, a general methodological challenge concerns the fact that this is only a small-scale feasibility study, where the number of participants was limited, and the duration of intervention was relatively short. These setups could influence the accuracy of our results. Additionally, due to the short duration, our RL model only used a pre-learned delivery for sending the reminders and was not able to adjust the strategy for each individual participant. Those reasons drive us to conduct a large-scale study with more participants and a longer intervention period. As examining the feasibility prior to large-scale effectiveness testing is an important preparation step in the development phase of a digital intervention [33], we think the study protocol and lessons learned in this feasibility study will be useful in the future study. Additionally, to examine the effectiveness of our reminder delivery function, we would like to cover a comparison between the adaptive timing and the non-adaptive timing in the large-scale study. Second, although we did not set up any gender bias during the participant recruitment, we are aware that most of our participants were female (accounting for 71.4%). Furthermore, due to the technical restriction, only people who have an Android smartphone could participate in this study. For those two reasons, it may be the case that our results are not generalized enough. We therefore would like to overcome this problem in the future by better balancing the gender of participants and expanding the possible users.

The lessons learned from this study also motivate us to improve our intervention design in future research. For instance, as shown from the user experience results, we still need to improve the delivery timing of our reminders for influencing the user’s future actions, which corresponds to the actionability of reminders [30]. As mentioned by some participants, factors such as the weather can be very effective. Previous JITAI studies also indicated that several factors (e.g., weather and personal mood) can play an important role in determining well personalized timing of reminders [5]. We therefore would like to take more momentary context information into account to improve the strategy of our reinforcement learning approach. Second, similar to the preferred timing, the preferred frequency of reminders is dramatically different among participants based on the questionnaire and interview results. For instance, some participants prefer 2 or 3 messages a week, while others would like to have two a day. We think this phenomenon will be taken into account in the future study. One possible solution is to give a different initialized strategy to the different user clusters based on their frequency preference [34].

## Figures and Tables

**Figure 1 ijerph-18-06059-f001:**
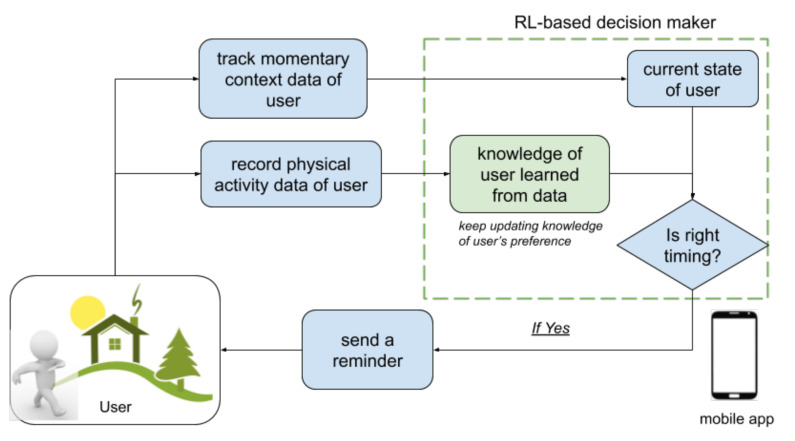
Overview of the reminder delivery procedure in the PAUL smartphone exercise application, regarding integration of a reinforcement learning model.

**Figure 2 ijerph-18-06059-f002:**
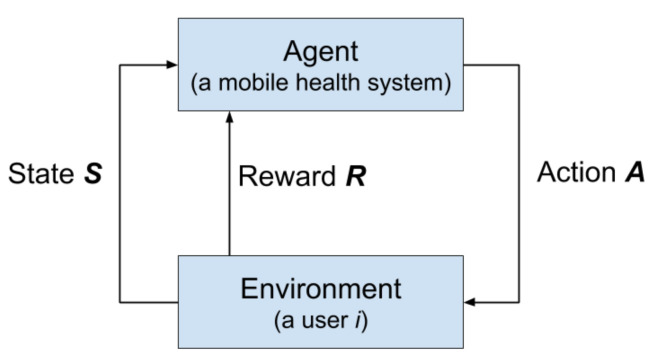
The interaction flow of our RL model, where the agent representing the smartphone health application and the environment is a user (Please note that the term reward is a specific entity in the MDP model. In our case, it indicates whether a physical activity is performed after the user receives a reminder sent by the RL model).

**Figure 3 ijerph-18-06059-f003:**
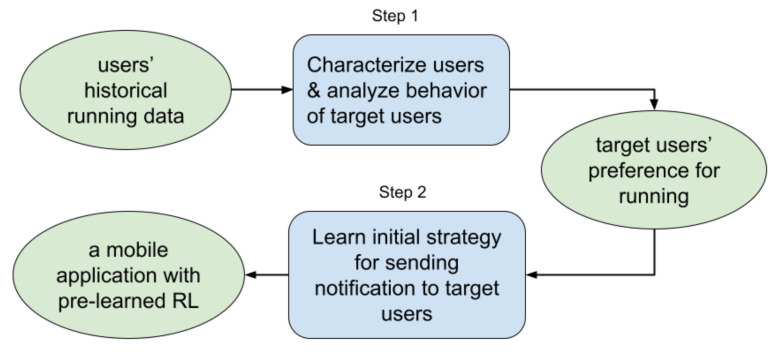
The overall framework for our pre-learned RL exercise application.

**Figure 4 ijerph-18-06059-f004:**
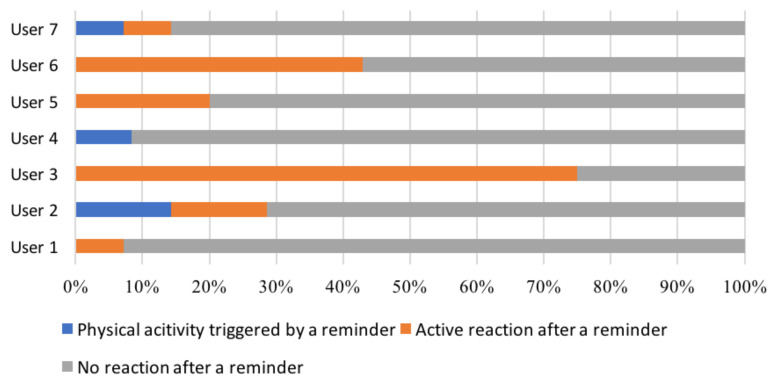
The reaction behaviors of all 7 users after receiving reminders from the PAUL app.

**Figure 5 ijerph-18-06059-f005:**
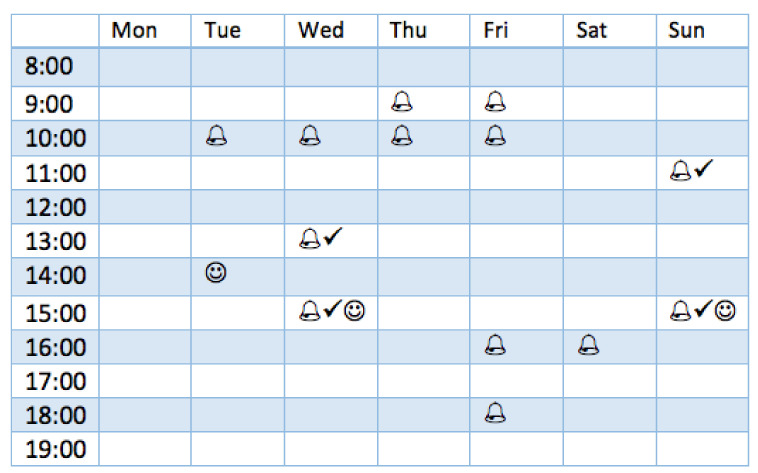
The reminder delivery and corresponding behaviors of user No. 2 in the intervention week, where a bell represents one reminder, a tick represents one short-term reaction (clicking the reminder pop-up), and a smile represents one physical activity.

**Figure 6 ijerph-18-06059-f006:**
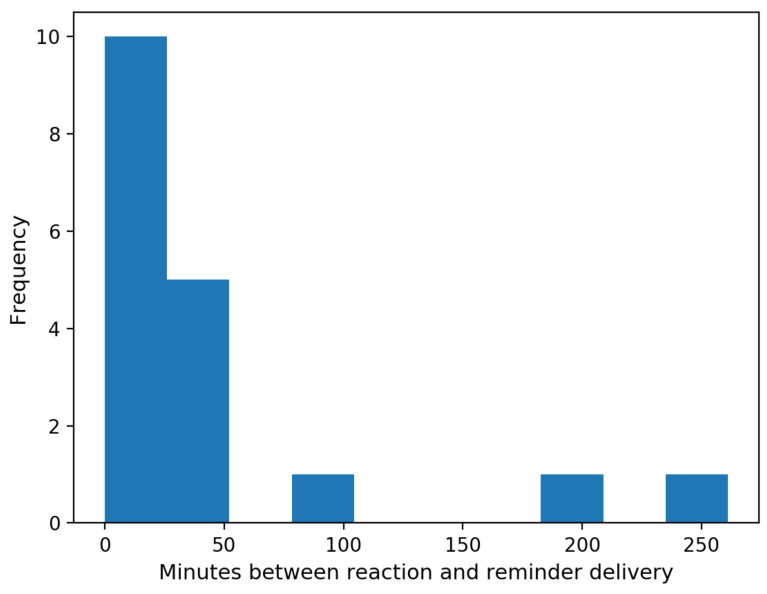
Distribution of the interval time between reminder delivery and reaction (when participants clicked a reminder).

**Figure 7 ijerph-18-06059-f007:**
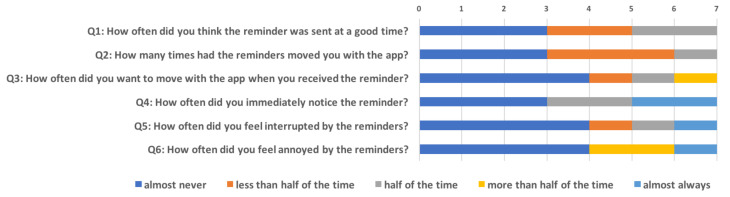
The results of the top 6 questions in the questionnaire, showing perceptions of the timing of reminders. Each unit on the *X* coordinate represents one participant.

**Table 1 ijerph-18-06059-t001:** The demographical information of 7 participants who completed the feasibility study.

Type	Category	Value
Gender		71.4% Female, 28.6% Male
Age		At average 34.4 (range from 21 to 51)
BMI		At average 24.73 (range from 19.72–36.99)
Residence	Near Utrecht Transwijk	3
	Near Amsterdam Oosterpark	3
	Near Amsterdam Sloterplas	1
Education level	Secondary school (VWO)	1
	Vocational education	1
	Associate degree	1
	University degree	4
Employment status	Part-time employment (<34 h/w)	3
	Full-time employment (≥34 h/w)	1
	Studying	3
Stage of change	Maintenance stage	4
(moderate activity)	Action stage	1
	Contemplation stage	2

**Table 2 ijerph-18-06059-t002:** Questions in the questionnaire and the interview about user experience on received reminders.

**Questions in the Questionnaire**
1. How often did you think the reminder was sent at a good time?
2. How many times had the reminders moved you with the app?
3. How often did you want to move with the app when you received the reminder?
4. How often did you immediately notice the reminder (when it had just been sent)?
5. How often did you feel interrupted by the reminders?
6. How often did you feel annoyed by the reminders?
7. Have you ever turned off the reminder function (by turning off the push notifications)?
8. What do you think of the number of reminders you received?
9. How much time would you like to have between the reminder and your exercise session?
**Topic list of the interview**
1. What do you think of the reminders sent by the app?
2. Did you find it useful? And why?
3. What did you think of the number of reminders?
What would be your ideal number of reminders?
4. Were you satisfied with the timing of the reminders?
Did the reminders bother you? Why? Do you have any suggestions for changing the timing?
5. Did the reminders motivate you to move more? How? Why or why not?
6. Can you think of something to improve the feature that we haven’t discussed yet?

**Table 3 ijerph-18-06059-t003:** Example data records indicate the behaviors of a user after receiving a reminder.

Event ID	User ID	Timestamp	Activity
3535	NO. 2	11 December 2019 15:00:00	reminder sent
3536	NO. 2	11 December 2019 15:04:00	notification clicked
3537	NO. 2	11 December 2019 15:04:01	app opened
3538	NO. 2	11 December 2019 15:07:00	walking or running session started

**Table 4 ijerph-18-06059-t004:** The results of other questions in the questionnaire about reminder-related properties.

Question	Choices	Value
Q7. Have you ever turned off the reminder function,	Yes, (almost) the entire intervention period	0
by turning off the push notification?	Yes, for multiple days	1
	Yes, but for a day or multiple hours	0
	No	6
Q8. What do you think of	way too few	2
the number of reminders you received?	too few	1
	good	2
	too much	2
	way too much	0
Q9. How much time would you like to have	less than 10 min	0
between the reminder and your exercise session?	less than 1 h	2
	about 1 h	4
	between 1 and 2 h	0
	longer than 2 h	1

## Data Availability

The datasets for this article are not publicly available because the collected data (e.g., application usage data, interviews) are privacy sensitive. No consent has been given to publicly share this data, but the data can be made available on request for verification purposes. Request to access the datasets should be directed to the corresponding author (s.wang2@uu.nl).

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
