# Peer review of "Reinforcement Learning to Send Reminders at Right Moments in Smartphone Exercise Application: A Feasibility Study"

_ijerph, 2021, doi:10.3390/ijerph18116059_

Round 1

Reviewer 1 Report

Congratulations to the authors for their work. Although one of the limitations of the study is the sample. The design and subject matter is very interesting.
Some improvements are recommended.
There is an error in figure 5. It appears as 2.png. I do not know if the colours of the figure are in accordance with the demands of the journal.
The discussion is too brief and the references used in the introduction are hardly used.
It would be interesting to clarify the conclusions and limitations of the study.

Author Response

Dear reviewer,

We appreciate your comments and the time you spend on reviewing our paper. The points you raised were very helpful for improving our manuscript. We address your comments point by point in the attached file.

Reviewer 2 Report

The aim of the study is interesting, however the article needs: 

a) Explain how the outputs support " Reinforcement Learning in concrete 

"b) Add some theoretical framework and show the consistency of the results with the theoretical framework 

c) Discuss bias in the selection of the participants. Only individuals positive to the app are included in the study. In addition to this the number of participants is very low. How pr if the number of participants influence the accuracy of the results have to be discussed 

d) Some gender issues ? most of the participants  are women.  

e) The description of the AAP seems to be a part of the results. 

d) What is the scientific contribution of the study? .

e) What kind of study has been performed. A case study, an evaluation study , other ? 

f) How data was analyzed? How the results were validated ? 

Author Response

(The authors gave the same response as above.)

Reviewer 3 Report

The authors (AA) aim to develop a reinforcement learning model in a smartphone exercise application and to evaluate this model in a feasibility study with real target users. This is an engaging article, but AA should rewrite manuscript taking into account the instructions for authors of IJERPH. The structure of manuscript should be: Introduction, Materials and Methods, Results, Discussion, Conclusions.

The manuscript is a little bit confusing and unclear. Methods and results are not clearly presented.

The title should clarify that this is a pilot study.

The abstract should be improved. It is not clear regarding what the study found (results and conclusions) and how they did it (methods). AA should report the main findings. AA report ten users but results are showed for 7 participants.

Line 11: Define RL.

In the introduction section, there is no explanation of the public health issue that makes necessary the development of this model. AA should better report what is already known about this topic. Moreover, the aims of the manuscript should be better outlined.

Lines 52-58: AA should eliminate this paragraph and use the structure defined by journal.

In the following sections (2-5), it is not clear where the background ends and the used methodology and the results begin. Was the questionnaire multiple choice? Likert scale?

Figure 2: it is not clear about reward.

The sample included is small to reach conclusions about the feasibility of the model. How do AA explain their choice?

AA should report this as a pilot study.

The results and discussion overlap in some parts (e.g. lines 265-266, 292-294, 304-305, etc.). The results are dispersive.

The discussion must be improved. AA should reorganize the discussion by making it less dispersive and discussing their results in light of existing literature. AA should start the discussion with their main findings. They should better define the study limitations and strengths.

AA should properly reorganize the manuscript before resubmitting it.

Author Response

(The authors gave the same response as above.)

Reviewer 4 Report

It is a good paper, but needs some polishing, check the attached document

well presented, i enjoyed reading it. Congratulations

Author Response

(The authors gave the same response as above.)

Round 2

Reviewer 3 Report

The authors (AA) have carefully addressed the reviewers' comments. Overall the changes made have improved the manuscript.

Accordingly, I would suggest acceptance after a minor revision.

In the abstract, AA should add the number of participants.

Lines 276-277 and 313-318: AA should move these sentences into the discussion. These sentences are not results.

Line 351: AA should delete the brackets: (or her).

Line 408: AA should delete this part.

Author Response

Dear reviewer,

We again appreciate your comments and the time you spend reviewing our paper. The points you raised were very helpful for improving our manuscript. We address your comments point by point in the attached PDF.
